# Group Fairness in Peer Review

**Haris Aziz**
UNSW Sydney
haris.aziz@unsw.edu.au

**Evi Micha**
University of Toronto
emicha@cs.toronto.edu

**Nisarg Shah**
University of Toronto
nisarg@cs.toronto.edu

## Abstract

Large conferences such as NeurIPS and AAAI serve as crossroads of various AI fields, since they attract submissions from a vast number of communities. However, in some cases, this has resulted in a poor reviewing experience for some communities, whose submissions get assigned to less qualified reviewers outside of their communities. An often-advocated solution is to break up any such large conference into smaller conferences, but this can lead to isolation of communities and harm interdisciplinary research. We tackle this challenge by introducing a notion of group fairness, called the core, which requires that every possible community (subset of researchers) to be treated in a way that prevents them from unilaterally benefiting by withdrawing from a large conference.

We study a simple peer review model, prove that it always admits a reviewing assignment in the core, and design an efficient algorithm to find one such assignment. We use real data from CVPR and ICLR conferences to compare our algorithm to existing reviewing assignment algorithms on a number of metrics.

## 1 Introduction

Due to their large scale, conferences like NeurIPS and AAAI use an automated procedure to assign submitted papers to reviewers. Popular such systems include the Toronto Paper Matching System [1], Microsoft CMT[1], and OpenReview[2]. The authors submitting their works are often very interested in receiving meaningful and helpful feedback from their peers [2–4]. Thus, their overall experience with the conference heavily depends on the quality of reviews that their submissions receive.

The typical procedure of assigning papers to reviewers is as follows. First, for each paper-reviewer pair, a similarity score is calculated based on various parameters such as the subject area of the paper and the reviewer, the bids placed by the reviewer, etc. [1, 5–8]. Then, an assignment is calculated through an optimization problem, where the usual objectives are to maximize either the utilitarian social welfare, which is the total similarity score of all matched paper-reviewer pairs, or the egalitarian social welfare, which is the least total score of reviewers assigned to any submission. Relevant constraints are imposed to ensure that each submission receives an appropriate number of reviews, reviewer workloads are respected, and any conflicts of interest are avoided.

Peng et al. [9] recently mentioned that a major problem with the prestigious mega conferences is that they constitute the main venues for several communities, and as a result, in some cases, people are asked to review submissions that are beyond their main areas of work. They claim that a reasonable solution is to move to a de-centralized publication process by creating more specialized conferences appropriate for different communities. While specialized conferences definitely have their advantages, the maintenance of large conferences that attract multiple communities is also crucial for the emergence of interdisciplinary ideas that can be reviewed by diverse subject experts. Therefore, it is important to ensure that no group has an incentive to break off due to a feeling of being mistreated by the reviewing procedure of a large conference. In this work, we ask whether it

---

[1] https://cmt3.research.microsoft.com/
[2] https://github.com/openreview/openreview-matcher

37th Conference on Neural Information Processing Systems (NeurIPS 2023).

is possible to modify the existing reviewing processes to resolve this issue by treating the various communities satisfactorily. We clarify that the goal is not to build roadblocks to the creation of specialized conferences, but rather to mitigate the harm imposed on communities in large conferences.

To answer this, we look toward the literature on algorithmic fairness. Specifically, we adapt the group fairness notion known as *the core* [10]; to the best of our knowledge, we are the first to introduce it to the peer review setting. For a reviewing assignment to be in the core, it must ensure that no community (subset of researchers) can "deviate" by setting up its own conference in which (a) no author reviews her own submission, (b) each submission from within the community is reviewed by just as many reviewers as before, but now from within the community, (c) each reviewer reviews no more papers than before, and (d) the submissions are assigned to better reviewers, making the community happier. Intuitively, this is a notion of group fairness because it ensure that the treatment provided to every group of participants meets their "entitlement" (which is defined by what the group can achieve on its own). It is also a notion of stability (often also known as *core stability*) because it provides no incentives for any community to break off and get isolated by setting up their own conference instead. Note that this definition provides fairness to *every possible* community, and not only to predefined groups, as is the case for fairness definitions such as demographic parity and equalized odds that are popular in the machine learning literature [11]. In particular, it ensures fair treatment to emerging interdisciplinary communities even before they become visible.

## 1.1 Our Contribution

We consider a simple peer review model in which each agent submits (as the sole author) a number of papers to a conference and also serves as a potential reviewer. A reviewing assignment is valid if each paper is reviewed by $k_p$ reviewers, each reviewer reviews no more than $k_a$ papers, and no agent reviews her own submissions. To ensure that a valid assignment always exists, we assume that the maximum number of papers that each agent is allowed to submit is at most $\lfloor k_a/k_p \rfloor$.

In Section 3, we present an efficient algorithm that always returns a valid assignment in the core under minor conditions on the preferences of the authors. Specifically, our algorithm takes as input only the preference ranking of each author over individual potential reviewers for each of her submissions. Then, it produces an assignment that we prove to be in the core for any manner in which the agent's submission-specific preferences over individual reviewers may be extended to preferences over a set of $k_p$ reviewers assigned to each submission, aggregated over submissions, subject to two mild conditions being satisfied.

In Section 4, we conduct experiments with real data from CVPR and ICLR conferences, and evaluate the price that our algorithm must pay — in lost utilitarian and egalitarian welfare — in order to satisfy the core and prevent communities from having an incentive to deviate. We also observe that reviewer assignment methods currently used in practice generate such adverse incentives quite often.

## 1.2 Related Work

As we mentioned above, usually the first step of a review assignment procedure is to calculate a similarity score for each pair of submission and reviewer which aims to capture the expertise of the reviewer for this submission. The problem of identifying the similarity scores has been extensively studied in the literature [1, 5–8, 12]. In this work, we assume that the similarity scores are given as an input to our algorithm after they have been calculated from a procedure that is considered as a black box. Importantly, our algorithm does not need the exact values of the similarity scores, but it only requires a ranking of the reviewers for each paper, indicating their relative expertise for this paper.

Given, the similarities scores various methods have been proposed for finding a reviewing assignment. The most famous algorithm is the Toronto Paper Matching System [1] which is a very broadly applied method and focuses on maximizing the utilitarian welfare, i.e. the sum of the similarities across all assigned reviewers and all papers. This approach has been adopted by other popular conference management systems such as EasyChair[3] and HotCRP [4] [13]. While this approach optimizes the total welfare, it is possible to discriminate against some papers. Therefore, other methods have focused on finding reviewing assignments that are (also) fair across all papers.

---

[3] https://easychair.org
[4] https://hotcrp.com

O'Dell et al. [14] suggest a method where the goal is to maximize the total score that a paper gets, while Stelmakh et al. [13] generalized this method by maximizing the minimum paper score, then maximizing the next smallest paper score, etc. Hartvigsen et al. [15] ensure fairness by requiring that each paper is assigned at least one qualified reviewer. Kobren et al. [16] proposed two algorithms that maximize that total utilitarian under the constraint that each paper should receive a score that exceeds a particular threshold. Payan and Zick [17] used the idea of envy-freeness [18] from the fair division literature to ensure fairness over the submissions. Moreover, some other works have focused on being fair over the reviewers rather than the papers [19, 20]). The core property that we consider in this work can be viewed as a fairness requirement over groups of authors. The reader can find more details about the challenges of the peer review problem in the recent survey of Shah [21].

In our model, the review assignment problem is related to exchange problems with endowments [22], since authors can be viewed as being endowed by their own papers which they wish to exchange with other authors that also serve as reviewers. For the basic exchange problem of housing reallocation, Shapley and Scarf [22] showed that an algorithm called *Top-Trading-Cycle (TTC)* finds an allocation which is in the core. The first part of our algorithm uses a variation of TTC where the agents (authors) are incorporated with multiple items (submissions), and constraints related to how many items each agent can get and to how many agents one item should be assigned should be satisfied. In contrast to classical exchange problem with endowments, our model has a distinctive requirement that agents/authors need to give away *all* their items/papers as the papers need to be reviewed by the agent who gets the paper. As we further explain in Section 3, this difference is crucial and requires further action from our algorithm than simply executing this variation of TTC. Various variations of TTC have been considered in the literature, tailored for different variations of the basic problem, but to the best of our knowledge, none of them can be directly applied in our model. To give an example, Suzuki et al. [23] consider the case that there are multiple copies of the same object and there are some quotas that should be satisfied, but they assume that each agent gets just one object while here each paper is assigned to multiple distinct reviewers.

## 2 Model

For $q \in \mathbb{N}$, define $[q] \triangleq \{1, \ldots, q\}$. There is a set of agents $N = [n]$. Each agent $i$ submits a set of papers $P_i = \{p_{i,1}, \ldots, p_{i,m_i}\}$ for review by her peers, where $m_i \in \mathbb{N}$, and is available to review the submissions of her peers. We refer to $p_{i,\ell}$ as the $\ell$-th submission of agent $i$; when considering the special case of each agent $i$ having a single submission, we will drop $\ell$ and simply write $p_i$. Let $P = \cup_{i \in N} P_i$ be the set of all submissions and $m = \sum_{i \in N} m_i$ be the total number of submissions.

**Assignment.** Our goal is to produce a *(reviewing) assignment* $R : N \times P \to \{0, 1\}$, where $R(i, j) = 1$ if agent $i \in N$ is assigned to review submission $j \in P$. With slight abuse of notation, let $R_i^a = \{j \in P : R(i, j) = 1\}$ be the set of submissions assigned to agent $i$ and $R_j^p = \{i \in N : R(i, j) = 1\}$ be the set of agents assigned to review submission $j$. We want the assignment to be *valid*, i.e., satisfy the following constraints:

- Each agent must be assigned at most $k_a$ submissions for review, i.e., $|R_i^a| \leqslant k_a, \forall i \in N$.
- Each submission must be assigned to $k_p$ agents, i.e., $|R_j^p| = k_p, \forall j \in P$.
- No agent should review one of her own submissions, i.e., $R(i, p_{i,\ell}) = 0, \forall i \in N, \ell \in [m_i]$.

To ensure that a valid assignment always exists, we impose the constraint that $m_i \cdot k_p \leqslant k_a$ for each $i \in N$, which implies that $m \cdot k_p \leqslant n \cdot k_a$. Intuitively, this demands that each agent submitting papers be willing to provide as many reviews as the number of reviews assigned to the submissions of any single agent. For further discussion on this condition, see Section 5.

Note that given $N' \subseteq N$ and $P_i' \subseteq P_i$ for each $i \in N'$ with $P' = \cup_{i \in N'} P_i'$, the validity requirements above can also be extended to a restricted assignment $\widehat{R} : N' \times P' \to \{0, 1\}$. Hereinafter, we will assume validity unless specified otherwise or during the process of building an assignment.

**Preferences.** Each agent $i \in N$ has a preference ranking, denoted $\sigma_{i,\ell}$, over the agents in $N \setminus \{i\}$ for reviewing her $\ell$-th submission $p_{i,\ell}$.[5] These preferences can be based on a mixture of many factors, such as how qualified the other agents are to review submission $p_{i,\ell}$, how likely they are to provide

---

[5]Our algorithms continue to work with weak orders; one can arbitrarily break ties to convert them into strict orders before feeding them to our algorithms.

a positive review for it, etc. Let $\sigma_{i,\ell}(i')$ be the position of agent $i' \in N \setminus \{i\}$ in the ranking. We say that agent $i$ prefers agent $i'$ to agent $i''$ as a reviewer for $p_{i,\ell}$ if $\sigma_{i,\ell}(i') < \sigma_{i,\ell}(i'')$. Again, in the special case where the agents have a single submission each, we drop $\ell$ and just write $\sigma_i$. Let $\vec{\sigma} = (\sigma_{1,1}, \ldots, \sigma_{1,m_1}, \ldots, \sigma_{n,1}, \ldots, \sigma_{n,m_n})$.

While our algorithm takes $\vec{\sigma}$ as input, to reason about its guarantees, we need to define agent preferences over assignments by extending $\vec{\sigma}$. In particular, an agent is assigned a set of reviewers for each of her submissions, so we need to define her preferences over sets of sets of reviewers. First, we extend to preferences over sets of reviewers for a given submission, and then aggregate preferences across different submissions. Instead of assuming a specific parametric extension (e.g., additive preferences), we allow all possible extensions that satisfy two mild constraints; the group fairness guarantee of our algorithm holds with respect to any such extension.

*Extension to a set of reviewers for one submission:* Let $S \succ_{i,\ell} S'$ (resp., $S \succcurlyeq_{i,\ell} S'$) denote that agent $i$ strictly (resp., weakly) prefers the set of agents $S$ to the set of agents $S'$ for her $\ell$-th submission $p_{i,\ell}$. We require only that these preferences satisfy the following mild axiom.

**Definition 1** (Order Separability). For every disjoint $S_1, S_2, S_3 \subseteq N$ with $|S_1| = |S_2| > 0$, if it holds that $\sigma_{i,\ell}(i') < \sigma_{i,\ell}(i'')$ for each $i' \in S_1$ and $i'' \in S_2$, then we must have $S_1 \cup S_3 \succ_{i,\ell} S_2 \cup S_3$.

An equivalent reformulation is that between any two sets of reviewers $S$ and $T$ with $|S| = |T|$, ignoring the common reviewers in $S \cap T$, if the agent strictly prefers every (even the worst) reviewer in $S \setminus T$ to every (even the best) reviewer in $T \setminus S$, then the agent must strictly prefer $S$ to $T$.

**Example 1.** Consider the common example of additive preferences, where each agent $i$ has a utility function $u_{i,\ell} : N \setminus \{i\} \to \mathbb{R}_{\geqslant 0}$ over individual reviewers for her $\ell$-th submission, inducing her preference ranking $\sigma_{i,\ell}$. In practice, these utilities are sometimes called similarity scores. Her preferences over sets of reviewers are defined via the additive utility function $u_{i,\ell}(S) \triangleq \sum_{i' \in S} u_{i,\ell}(i')$. It is easy to check that for any disjoint $S_1, S_2, S_3$ with $|S_1| = |S_2| > 0$, $u_{i,\ell}(i') > u_{i,\ell}(i'')$ for all $i' \in S_1$ and $i'' \in S_2$ would indeed imply $u_{i,\ell}(S_1 \cup S_3) > u_{i,\ell}(S_2 \cup S_3)$. Additive preferences are just one example from a broad class of extensions satisfying order separability.

*Extension to assignments.* Let us now consider agent preferences over sets of sets of reviewers, or equivalently, over assignments. Let $R \succ_i \widehat{R}$ (resp., $R \succcurlyeq_i \widehat{R}$) denote that agent $i$ strictly (resp., weakly) prefers assignment $R$ to assignment $\widehat{R}$. Note that these preferences collate the submission-wise preferences $\succ_{i,\ell}$ across all submissions of the agent. We require only that the preference extension satisfies the following natural property.

**Definition 2** (Consistency). For any assignment $R$, restricted assignment $\widehat{R}$ over any $N' \subseteq N$ and $P' = \cup_{i \in N'} P'_i$ (where $P'_i \subseteq P_i$ for each $i \in N'$), and agent $i^* \in N'$, if it holds that $R^p_{p_{i^*,\ell}} \succcurlyeq_{i^*,\ell} \widehat{R}^p_{p_{i^*,\ell}}$ for each $p_{i^*,\ell} \in P'_i$, then we must have $R \succcurlyeq_i \widehat{R}$.

In words, if an agent weakly prefers $R$ to $\widehat{R}$ for the set of reviewers assigned to each of her submissions individually, then she must prefer $R$ to $\widehat{R}$ overall.

**Example 2.** Let us continue with the previous example of additive utility functions. The preferences of agent $i$ can be extended additively to assignments using the utility function $u_i(R) = \sum_{p_{i,\ell} \in P} u_{i,\ell}(R^p_{p_{i,\ell}})$. It is again easy to check that if $u_{i,\ell}(R^p_{p_{i,\ell}}) \geqslant u_{i,\ell}(\widehat{R}^p_{p_{i,\ell}})$ for each $p_{i,\ell}$, then $u_i(R) \geqslant u_i(\widehat{R})$. Hence, additive preferences are again one example out of a broad class of preference extensions that satisfy consistency.

**Core.** Our goal is to find a group-fair assignment which treats every possible group of agents at least as well as they could be on their own, thus ensuring that no subset of agents has an incentive to deviate and set up their own separate conference. Formally:

**Definition 3** (Core). An assignment $R$ is in the core if there is no $N' \subseteq N$, $P'_i \subseteq P_i$ for each $i \in N'$, and restricted assignment $\widehat{R}$ over $N'$ and $P' = \cup_{i \in N'} P'_i$ such that $\widehat{R} \succ_i R$ for each $i \in N'$.

In words, if any subset of agents deviate with any subset of their submissions and implement any restricted reviewing assignment, at least one deviating agent would not be strictly better off, thus eliminating the incentive for such a deviation. We also remark that our algorithm takes only the preference rankings over individual reviewers $\vec{\sigma}$ as input and produces an assignment $R$ that is

---

**ALGORITHM 1:** CoBRA

**Input:** $N, P, \vec{\sigma}, k_a, k_p$
**Output:** $R$

1   $R, L, U = $PRA-TTC$(N, P, \vec{\sigma}, k_a, k_p)$;
2   **if** $|U| > 0$ **then**
3      |   $R = $Filling-Gaps$(N, P, \vec{\sigma}, k_a, k_p, R, L, U)$;

---

**ALGORITHM 2:** PRA-TTC

**Input:** $N, P, \vec{\sigma}, k_a, k_p$
**Output:** $R, L, U$

1   $R(i, j) \leftarrow 0, \forall i \in N$ and $\forall j \in P$;
2   Construct the preference graph $G_R$;
3   **while** $\exists$ *cycle in* $G_R$ **do**
4      |   Eliminate the cycle;
5      |   Update $\overline{P}_i$-s by removing any completely assigned paper;
6      |   Update $G_R$;
7   $U \leftarrow \{i \in N : \overline{P}_i \neq \emptyset\}$ ;
8   $L \leftarrow$ the last $k_p - |U| + 1$ agents in $N \setminus U$ to have all their submissions completely assigned ;

---

guaranteed to be in the core according to *every preference extension* of $\vec{\sigma}$ satisfying order separability and consistency.

## 3   CoBRA: An Algorithm for Computing Core-Based Reviewer Assignment

In this section, we prove our main result: when agent preferences are order separable and consistent, an assignment in the core always exists and can be found in polynomial time.

**Techniques and key challenges:** The main algorithm CoBRA (Core-Based Reviewer Assignment), presented as Algorithm 1, uses two other algorithms, PRA-TTC and Filling-Gaps, presented as Algorithm 2 and Algorithm 3, respectively. We remark that PRA-TTC is an adaptation of the popular Top-Trading-Cycles (TTC) mechanism, which is known to produce an assignment in the core for the house reallocation problem (and its variants) [22]. The adaptation mainly incorporates the constraints related to how many papers each reviewer can review and how many reviewers should review each paper. While for $k_p = k_a = 1$, PRA-TTC is identical with the classic TTC that is used for the house reallocation problem, the main difference of this problem with the review assignment problem is that in the latter each agent should give away her item (i.e. her submission) and obtain the item of another agent. Therefore, by simply executing TTC in the review assignment problem, one can get into a deadlock before producing a valid assignment. For example, consider the case of three agents, each with one submission. Each submission must receive one review ($k_p = 1$) and each agent provides one review ($k_a = 1$). The TTC mechanism may start by assigning agents 1 and 2 to review each other's submission, but this cannot be extended into a valid assignment because there is no one left to review the submission of agent 3. This is where Filling-Gaps comes in; it makes careful edits to the partial assignment produced by the PRA-TTC, and the key difficulty is to prove that this produces a valid assignment while still satisfying the core.

### 3.1   Description of CoBRA

Before we describe CoBRA in detail, let us introduce some more notation. Let $m^* = \max_{i \in N} m_i$. For reasons that will become clear later, we want to ensure that $m_i = m^*$, for each $i \in N$. To achieve that, we add $m^* - m_i$ dummy submissions to agent $i$, and the rankings over reviewers with respect to these submissions are arbitrarily. An assignment is called *partial* if there are submissions that are reviewed by less than $k_p$ agents. A submission that is reviewed by $k_p$ agents under a partial assignment is called *completely assigned*. Otherwise, it is called *incompletely assigned*. We denote with $\overline{P}_i(\widehat{R})$ the set of submissions of $i$ that are incompletely assigned under a partial assignment $\widehat{R}$. We omit $\widehat{R}$ from the notation when it is clear from context.

CoBRA calls PRA-TTC, and then if needed, it calls Filling-Gaps. Below, we describe the algorithms.

**ALGORITHM 3:** Filling-Gaps

**Input:** $N, P, \vec{\sigma}, k_a, k_p, R, L, U$
**Output:** $R$
**Phase 1**;
1 Construct the greedy graph $G_R$;
2 **while** $\exists$ *cycle* **do**
3      Eliminate the cycle ;
4      Update $\overline{P}_i$-s by removing any completely assigned paper;
5      Update $U$ and $L$ by moving any agent $i$ from $U$ to $L$ if $\overline{P}_i = \emptyset$;
6      Update $G_R$;
**Phase 2**;
7 Construct the topological order $\vec{\rho}$ of $G_R$;
8 **for** $t \in [|U|]$ **do**
9      **while** $\overline{P}_{\rho(t)} \neq \emptyset$ **do**
10          Pick arbitrary $p_{\rho(t),\ell} \in \overline{P}_{\rho(t)}$ ;
11          Find completely assigned $p_{i',\ell'}$, with $R(\rho(t), p_{i',\ell'}) = 0$, for some $i' \in U \cup L \setminus \{\rho(t)\}$ ;
12          Find $i'' \neq \rho(t)$ such that $R(i'', p_{\rho(t),\ell}) = 0$ and $R(i'', p_{i',\ell'}) = 1$ ;
13          $R(i'', p_{\rho(t),\ell}) \leftarrow 1; R(i'', p_{i',\ell'}) \leftarrow 0; R(\rho(t), p_{i',\ell'}) \leftarrow 1$ ;
14          Remove $p_{\rho(t),\ell}$ from $\overline{P}_{\rho(t)}$ if it is completely assigned;

**PRA-TTC.** In order to define PRA-TTC, we first need to introduce the notion of a *preference graph*. Suppose we have a partial assignment $\widehat{R}$. Each agent $i$ with $\overline{P}_i \neq \emptyset$ picks one of her incompletely assigned submissions arbitrarily. Without loss of generality, we assume that she picks her $\ell^*$-th submission. We define the directed preference graph $G_{\widehat{R}} = (N, E_{\widehat{R}})$ where each agent is a node and for each $i$ with $\overline{P}_i \neq \emptyset$, $(i, i') \in E_{\widehat{R}}$ if and only if $i'$ is ranked highest in $\sigma_{i,\ell^*}$ among the agents that don't review $p_{i,\ell^*}$ and review less than $k_a$ submissions. Moreover, for each $i \in N$ with $\overline{P}_i = \emptyset$, we add an edge from $i$ to $i'$, where $i'$ is an arbitrary agent with $\overline{P}_{i'} \neq \emptyset$. PRA-TTC starts with an empty assignment, constructs the preference graph and searches for a directed cycle. If such a cycle exists, the algorithm eliminates it as following: For each $(i, i')$ that is included in the cycle, it assigns submission $p_{i,\ell^*}$ to $i'$ (if $i$'s submissions are already completely assigned, it does nothing) and removes $p_{i,\ell^*}$ from $\overline{P}_i$, if it is now completely assigned. Then, the algorithm updates the preference graph and continues to eliminate cycles in the same way. When there are no left cycles in the preference graph, the algorithm terminates and returns two sets, $U$ and $L$. The first set contains all the agents that some of their submissions are incompletely assigned and the set $L$ contains the last $k_p - |U| + 1$ agents whose all submissions became completely assigned.

**Filling-Gaps.** CoBRA calls Filling-Gaps, if the $U$ that returned from PRA-TTC is non empty. Before we describe the Filling-Gaps, we also need to introduce the notion of a *greedy graph*. Suppose that we have a partial assignment $\widehat{R}$ which indicates a set $U$ that contains all the agents whose at least one submission is incompletely assigned. We define the directed greedy graph $G_{\widehat{R}} = (U, E_{\widehat{R}})$ where $(i, i') \in E_{\widehat{R}}$ if $\widehat{R}(i', p_{i,\ell}) = 0$ for some $p_{i,\ell} \in \overline{P}_i$. In other words, while in the preference graph, agent $i$ points only to her favourite potential reviewer with respect to one of her incomplete submissions, in the greedy graph agent $i$ points to any agent in $U \setminus \{i\}$ that could review at least one of her submissions that is incompletely assigned. Filling-Gaps consists of two phases. In the first phase, starting from the partial assignment $R$ that was created from PRA-TTC, it constructs the greedy graph, searches for cycles and eliminates a cycle by assigning $p_{i,\ell}$ to agent $i'$ for each $(i, i')$ in the cycle that exists due to $p_{i,\ell}$ (when an edge exists due to multiple submissions, the algorithm chooses one of them arbitrary). Then, it updates $\overline{P}_i$ by removing any $p_{i,\ell}$ that became completely assigned and also updates $U$ by moving any $i$ to $L$ if $\overline{P}_i$ became empty. It continues by updating the greedy graph and eliminating cycles in the same way. When no more cycles exist in the greedy graph, the algorithm proceeds to the second phase, where in $|U|$ rounds ensures that the incomplete submissions of each agent become completely assigned.

In the appendix, there is an execution of CoBRA in a small instance.

## 3.2 Main Result

We are now ready to present our main result.

**Theorem 1.** *When agent preferences are order separable and consistent, CoBRA returns an assignment in the core in $O(n^3)$ time complexity.*

*Proof.* First, in the next lemma, we show that CoBRA returns a valid assignment. The proof is quite non trivial and several pages long, so we defer it to the supplementary material.

**Lemma 1.** *CoBRA returns a valid assignment.*

Now, we show that the final assignment $R$ that CoBRA returns is in the core. Note that while it is possible that an assignment of a submission of an agent in $U \cup L$, that was established during the execution of PRA-TTC, to be removed in the execution of Filling-Gaps, this never happens for submissions that belong to some agent in $N \setminus (U \cup L)$. For the sake of contradiction, assume that $N' \subseteq N$, with $P'_i \subseteq P_i$ for each $i \in N'$, deviate to a restricted assignment $\widetilde{R}$ over $N'$ and $\cup_{i \in N'} P'_i$. Note that $\widetilde{R}$ is valid only if $|N'| > k_p$, as otherwise there is no way each submission in $\cup_{i \in N'} P'_i$ to be completely assigned, since no agent can review her own submissions.

We distinguish into two cases and we show that in both cases the assignment is in the core.

**Case I:** $\exists i \in N' : i \notin L \cup U$**.** Let $i^* \in N'$ be the first agent in $N'$ whose all submissions became completely assigned in the execution of PRA-TTC. Note that since there exists $i \notin U \cup L$, we get that $i^* \notin U \cup L$ from the definitions of $U$ and $L$. Now, consider any $p_{i^*,\ell}$. Let $Q_1 = R^p_{p_{i^*,\ell}} \setminus (R^p_{p_{i^*,\ell}} \cap \widetilde{R}^p_{p_{i^*,\ell}})$ and $Q_2 = \widetilde{R}^p_{p_{i^*,\ell}} \setminus (R^p_{p_{i^*,\ell}} \cap \widetilde{R}^p_{p_{i^*,\ell}})$. If $Q_1 = \emptyset$, then we have that $R^p_{p_{i^*,\ell}} = \widetilde{R}^p_{p_{i^*,\ell}}$ which means that $R^p_{p_{i^*,\ell}} \succcurlyeq_{i^*,\ell} \widetilde{R}^p_{p_{i^*,\ell}}$. Otherwise, let $i' = \text{argmax}_{i \in Q_1} \sigma_{i^*,\ell}(i)$, i.e. $i'$ is ranked at the lowest position in $\sigma_{i^*,\ell}$ among the agents that review $p_{i^*,\ell}$ under $R$ but not under $\widetilde{R}$. Moreover, let $i'' = \text{argmin}_{i \in Q_2} \sigma_{i^*,\ell}(i)$, i.e. $i''$ is ranked at the highest position in $\sigma_{i^*,\ell}$ among the agents that review $p_{i^*,\ell}$ under $\widetilde{R}$ but not under $R$. We have $R(i', p_{i^*,\ell}) = 1$, if and only if $i^*$ has an outgoing edge to $i'$ at some round of PRA-TTC. At the same round, we get that $i''$ can review more submissions, since $i'' \in N'$ and if $i^*$ has incompletely assigned submissions, then any $i \in N'$ has incompletely assigned submissions, and hence $|R^a_{i''}| < k_p \cdot m^* \leqslant k_a$. This means that if $\sigma_{i^*,\ell}(i') > \sigma_{i^*,\ell}(i'')$, then $i^*$ would point $i''$ instead of $i'$. We conclude that $\sigma_{i^*,\ell}(i') < \sigma_{i^*,\ell}(i'')$. Then, from the definition of $i'$ and $i''$ and from the order separability property we have that $R^p_{p_{i^*,\ell}} \succ_{i^*,\ell} \widetilde{R}^p_{p_{i^*,\ell}}$. Thus, either if $Q_1$ is empty or not, we have that for any $p_{i^*,\ell} \in P'_i$, it holds that $R^p_{p_{i^*,\ell}} \succcurlyeq_{i^*,\ell} \widetilde{R}^p_{p_{i^*,\ell}}$ and from consistency, we get that $R \succcurlyeq_{i^*} \widetilde{R}$ which is a contradiction.

**Case II:** $\nexists i \in N' : i \notin L \cup U$**.** In this case we have that $N' = U \cup L$, as $|U \cup L| = k_p + 1$. This means that for each $i \in U \cup L$ and $\ell \in [m^*]$, $\widetilde{R}^p_{p_{i,\ell}} = (U \cup L) \setminus \{i\}$. Let $i^* \in L$ be the first agent in $L$ whose all submissions became completely assigned in the execution of PRA-TTC. Consider any $p_{i^*,\ell}$. Note that it is probable that while $p_{i^*,\ell}$ was assigned to some agent $i$ in PRA-TTC, it was moved to another agent $i'$ during the execution of Filling-Gaps. But, $i'$ belongs to $U$ and we can conclude that if $p_{i^*,\ell}$ is assigned to some $i \in N \setminus U$ at the output of CoBRA, this assignment took place during the execution of PRA-TTC. Now, let $Q_1 = R^p_{p_{i^*,\ell}} \setminus (R^p_{p_{i^*,\ell}} \cap \widetilde{R}^p_{p_{i^*,\ell}})$ and $Q_2 = \widetilde{R}^p_{p_{i^*,\ell}} \setminus (R^p_{p_{i^*,\ell}} \cap \widetilde{R}^p_{p_{i^*,\ell}})$. If $Q_1 = \emptyset$, then we have that $R^p_{p_{i^*,\ell}} = \widetilde{R}^p_{p_{i^*,\ell}}$ which means that $R^p_{p_{i^*,\ell}} \succcurlyeq_{i^*,\ell} \widetilde{R}^p_{p_{i^*,\ell}}$. If $Q_1 \neq \emptyset$, then $Q_1 \subseteq N \setminus (U \cup L)$ and $Q_2 \subseteq U \cup L$ since $\widetilde{R}^p_{p_{i^*,\ell}} = U \cup L$. Let $i' = \text{argmax}_{i \in Q_1} \sigma_{i^*,\ell}(i)$, i.e. $i'$ is ranked at the lowest position in $\sigma_{i^*,\ell}$ among the agents that review $p_{i^*,\ell}$ under $R$ but not under $\widetilde{R}$. Moreover, let $i'' = \text{argmin}_{i \in Q_2} \sigma_{i^*,\ell}(i)$, i.e. $i''$ is ranked at the highest position in $\sigma_{i^*,\ell}$ among the agents that review $p_{i^*,\ell}$ under $\widetilde{R}$ but not under $R$. From above, we know that the assignment of $p_{i^*,\ell}$ to $i'$ was implemented during the execution of PRA-TTC, since $i' \in N \setminus (U \cup L)$. Hence, with very similar arguments as in the previous case, we will conclude that $\sigma_{i^*,\ell}(i') < \sigma_{i^*,\ell}(i'')$. We have $R(i', p_{i^*,\ell}) = 1$ if and only if $i^*$ has an outgoing edge to $i'$ at some round of PRA-TTC. At this round, we know that $i''$ can review more submissions, since $i'' \in N'$ and if $i^*$ has incompletely assigned submissions, then any $i \in N'$ has incompletely assigned submissions. This means that if $\sigma_{i^*,\ell}(i') > \sigma_{i^*,\ell}(i'')$, then $i^*$ would point $i''$ instead of $i'$. Hence, we conclude that

$\sigma_{i^*,\ell}(i') < \sigma_{i^*,\ell}(i'')$. Then, from the definition of $i'$ and $i''$ and from the order separability property we have that $R^p_{p_{i^*,\ell}} \succ_{i^*,\ell} \widetilde{R}^p_{p_{i^*,\ell}}$. Thus, either if $Q_1$ is empty or not, we have that for any $p_{i^*,\ell} \in P'_i$, it holds that $R^p_{p_{i^*,\ell}} \succcurlyeq_{i^*,\ell} \widetilde{R}^p_{p_{i^*,\ell}}$ and from consistency we get that $R \succcurlyeq_{i^*} \widetilde{R}$ which is a contradiction.

Lastly, we analyze the time complexity of CoBRA. First, we consider the time complexity of PRA-TTC. In each iteration, the algorithm assigns at least one extra reviewer to at least one incompletely-assigned submission. This can continue for at most $m \cdot k_p \leqslant n \cdot k_a$ iterations, since each submission should be reviewed by $k_p$ reviewers. In each iteration, it takes $O(n)$ time to find and eliminate a cycle in the preference graph. Then, it takes $O(n^2)$ time to update the preference graph, since for each arbitrarily-picked incompletely-assigned submission of each agent, we need to find the most qualified reviewer who can be additionally assigned to it. By all the above, we conclude that the runtime of PRA-TTC is $O(n^3)$, by ignoring $k_a$ which is a small constant in practice. After PRA-TTC terminates, CoBRA calls the Filling-Gaps algorithm. However, Lemma 3 ensures that at the end of PRA-TTC, $|L \cup U| \leqslant k_p + 1$, which is also a small constant. And Filling-Gaps only makes local changes that affect these constantly many agents. As such, the running time of Filling-Gaps is constant as well. Therefore, the time complexity of CoBRA is $O(n^3)$

$\square$

## 4   Experiments

In this section, we empirically compare CoBRA to TPMS [1], which is widely used (for example, it was used by NeurIPS for many years), and PR4A [13], which was used in ICML 2020 [24]. As mentioned in the introduction, these algorithms assume the existence of a similarity or affinity score for each pair of reviewer $i$ and paper $j$, denoted by $S(i, j)$. The score (or utility) of a paper under an assignment $R$, denoted by $u^p_j$, is computed as $u^p_j \triangleq \sum_{i \in R^p_j} S(i, j)$. TMPS finds an assignment $R$ that maximizes the utilitarian social welfare (USW), i.e., the total paper score $\sum_{j \in P} u^p_j$, whereas PR4A finds an assignment that maximizes the egalitarian social welfare (ESW), i.e., the minimum paper score $\min_{j \in P} u^p_j$.[6] We use $k_a = k_p = 3$ in these experiments.[7]

**Datasets.** We use three conference datasets: from the Conference on Computer Vision and Pattern Recognition (CVPR) in 2017 and 2018, which were both used by Kobren et al. [16], and from the International Conference on Learning Representations (ICLR) in 2018, which was used by Xu et al. [25]. In the ICLR 2018 dataset, similarity scores and conflicts of interest are also available. While a conflict between a reviewer and a paper does not necessarily indicate authorship, it is the best indication we have available, so, following Xu et al. [25], we use the conflict information to deduce authorship. Since in our model each submission has one author, and no author can submit more than $\lfloor k_a/k_p \rfloor = 1$ papers, we compute a maximum cardinality matching on the conflict matrix to find author-paper pairs, similarly to what Dhull et al. [26] did. In this way, we were able to match 883 out of the 911 papers. We disregard any reviewer who does not author any submissions, but note that the addition of more reviewers can only improve the results of our algorithm since these additional reviewers have no incentive to deviate. For the CVPR 2017 and CVPR 2018 datasets, similarity scores was available, but not the conflict information. In both these datasets, there are fewer reviewers than papers. Thus, we constructed artificial authorship relations by sequentially processing papers and matching each paper to the reviewer with the highest score for it, if this reviewer is still unmatched. In this way, we were able to match 1373 out of 2623 papers from CVPR 2017 and 2840 out of 5062 papers from CVPR 2018. In the ICLR 2018 and CVPR 2017 datasets, the similarity scores take values in $[0, 1]$, so we accordingly normalized the CVPR 2018 scores as well.

**Measures.** We are most interested in measuring the extent to which the existing algorithms provide incentives for communities of researchers to deviate. To quantify this, we need to specify the utilities of the authors. We assume that they are additive, i.e., the utility of each author in an assignment is the total similarity score of the $k_p = 3$ reviewers assigned to their submission.

---

[6]Technically, subject to this, it maximizes the second minimum paper score, and then the third minimum paper score, etc. This refinement is also known as leximin in the literature.

[7]Note that $k_p = 3$ reviews per submission is quite common, although the reviewer load $k_a$ is typically higher in many conferences, often closer to 6. However, the differences between different algorithms diminish with higher values of $k_a$.

| Dataset | Alg | USW | ESW | α-Core | | CV-Pr |
| | | | | #unb-α | α* | |
| --- | --- | --- | --- | --- | --- | --- |
| CVPR '17 | CoBRA | $1.225 \pm 0.021$ | $0.000 \pm 0.000$ | 0% | $1.000 \pm 0.000$ | 0% |
| | TPMS | $1.497 \pm 0.019$ | $0.000 \pm 0.000$ | 89% | $3.134 \pm 0.306$ | 100% |
| | PR4A | $1.416 \pm 0.019$ | $0.120 \pm 0.032$ | 51% | $1.700 \pm 0.078$ | 100% |
| CVPR '18 | CoBRA | $0.224 \pm 0.004$ | $0.004 \pm 0.001$ | 0% | $1.000 \pm 0.000$ | 0% |
| | TPMS | $0.286 \pm 0.005$ | $0.043 \pm 0.004$ | 0% | $1.271 \pm 0.038$ | 100% |
| | PR4A | $0.282 \pm 0.005$ | $0.099 \pm 0.001$ | 0% | $1.139 \pm 0.011$ | 100% |
| ICLR '18 | CoBRA | $0.166 \pm 0.001$ | $0.028 \pm 0.001$ | 0% | $1.000 \pm 0.000$ | 0% |
| | TPMS | $0.184 \pm 0.001$ | $0.048 \pm 0.002$ | 0% | $1.048 \pm 0.008$ | 90% |
| | PR4A | $0.179 \pm 0.001$ | $0.082 \pm 0.001$ | 0% | $1.087 \pm 0.009$ | 100% |

Table 1: Results on CVPR 2017 and 2018, and ICLR 2018.

*Core violation factor:* Following the literature [27], we measure the multiplicative violation of the core (if any) that is incurred by TPMS and PR4A (CoBRA provably does not incur any). This is done by computing the maximum value of $\alpha \geqslant 1$ for which there exists a subset of authors such that by deviating and implementing some valid reviewing assignment of their papers among themselves, they can each improve their utility by a factor of at least $\alpha$. This can easily be formulated as a binary integer linear program (BILP). Because this optimization is computationally expensive (the most time-consuming component of our experiments), we subsample $100$ papers[8] from each dataset in each run, and report results averaged over 100 runs. Note that whenever there exists a subset of authors with zero utility each in the current assignment who can deviate and receive a positive utility each, the core deviation $\alpha$ becomes infinite. We separately measure the percentage of runs in which this happens (in the column #unb-$\alpha$), and report the average $\alpha$ among the remaining runs in the $\alpha^*$ column.

*Core violation probability:* We also report the percentage of runs in which a core violation exists (i.e., there exists at least one subset of authors who can all strictly improve by deviating from the current assignment). We refer to this as the *core violation probability* (CV-Pr).

*Social welfare:* Finally, we also measure the utilitarian and egalitarian social welfare (USW and ESW) defined above, which are the objectives maximized by TPMS and PR4A, respectively.

**Results.** The results are presented in Table 1. As expected, TPMS and PR4A achieve the highest USW and ESW, respectively, on all datasets because they are designed to optimize these objectives. In CVPR 2017, CoBRA and TPMS always end up with zero ESW because this dataset includes many zero similarity scores, but PR4A is able to achieve positive ESW. In all datasets, CoBRA achieves a relatively good approximation with respect to USW, but this is not always the case with respect to ESW. For example, in CVPR 2018, CoBRA achieves 0.004 ESW on average whereas PR4A achieves 0.099 ESW on average. This may be due to the fact that this dataset also contains many zero similarity scores, and the myopic process of CoBRA locks itself into a bad assignment, which the global optimization performed by PR4A avoids.

While CoBRA suffers some loss in welfare, TPMS and PR4A also generate significant adverse incentives. They incentivize at least one community to deviate in almost every run of each dataset (CV-Pr). While the magnitude of this violation is relatively small when it is finite (except for TPMS in CVPR 2017), TPMS and PR4A also suffer from unbounded core violations in more than half of the runs for CVPR 2017; this may again be due to the fact that many zero similarity scores lead to deviations by groups where each agent has zero utility under the assignments produced by these algorithms.

Of all these results, the high probability of core violation under TPMS and PR4A is perhaps the most shocking result; when communities regularly face adverse incentives, occasional deviations may happen, which can endanger the stability of the conference. That said, CoBRA resolves this issue at a significant loss in fairness (measured by ESW). This points to the need for finding a middle ground where adverse incentives can be minimized without significant loss in fairness or welfare.

---

[8]In Table 2 in the appendix, we report USW and ESW without any subsampling and we note that the qualitative relationships between the different algorithms according to each metric remain the same.

# 5 Discussion

In this work, we propose a way for tackling the poor reviewing problem in large conferences by introducing the concept of "core" as a notion of group fairness in the peer review process. This fairness principle ensures that each subcommunity is treated at least as well as it would be if it was not part of the larger conference community.

We show that under certain —albeit sometimes unrealistic—assumptions , a peer review assignment in the core always exists and can be efficiently found. In the experimental part, we provide evidence that peer review assignment procedures that are currently used in practice, quite often motivate subcommunities to deviate and build their own conferences.

Our theoretical results serve merely as the first step toward using it to find reviewer assignments that treat communities fairly and prevent them from deviating. As such, our algorithm has significant limitations that must be countered before it is ready for deployment in practice. A key limitation is that it only works for single-author submissions, which may be somewhat more realistic for peer review of grant proposals, but unrealistic for computer science conferences. We also assume that each author serves as a potential reviewer; while many conferences require this nowadays, exceptions must be allowed in special circumstances. We also limit the number of submissions by any author to be at most $\lfloor k_a/k_p \rfloor$, which is a rather small value in practice, and some authors ought to submit more papers than this. We need to make this assumption to theoretically guarantee that a valid assignment exists. An interesting direction is to design an algorithm that can produce a valid assignment in the (approximate) core whenever it exists. Finally, deploying group fairness in real-world peer review processes may require designing algorithms that satisfy it approximately at minimal loss in welfare, as indicated by our experimental results.

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
