# OpenReview forum: "Group Fairness in Peer Review"
_NeurIPS.cc/2023/Conference — NeurIPS 2023 spotlight_

### Official Review · Reviewer_wnxf · 2023-07-05

**Soundness:** 3 good
**Presentation:** 2 fair
**Contribution:** 2 fair
**Rating:** 6
**Confidence:** 3

**Summary:**

The paper describes a new algorithm for assigning papers to peer reviewers with the goal of ensuring that review assignments are in the core. Thie motivation behind this is that large conferences benefit from joining many different communities of research and thereby enabling more interdisciplinary cooperation. If review assignments are in the core then subgroups will have no incentive to break off and form smaller, more siloed conferences.

An algorithm is described that, under certain assumptions, creates assignments between papers and authors that are within the core and claims to run in polynomial time. This algorithm is compared with two commonly used algorithms on a number of metrics. While CoBRA, the new algorithm, does not provide as much utility as the TPMS and PR4A algorithms it is shown that the other algorithms almost always produce outcomes that are not in the core. The paper concludes by suggesting that some middle ground between these approaches may be a useful path forward.

**Strengths:**

While the problem described by the paper has received a fair bit of research the application of the core appears to be a novel approach to the review assignment problem. The problem is not of critical importance but is one that has room for improvement and can relatively easily benefit from algorithmic enhancements.

Overall the paper provides clear high-level descriptions of each step and the motivation behind why the authors believe the core is a useful idea for review assignment. The authors are also clear about the limitations of their work and where it does and does not improve upon the status quo.

The experiment section was a useful addition. Despite the results not dominating other algorithms I am glad there is some evaluation of CoBRA.

**Weaknesses:**

While the high level components of the paper are explained quite clearly I found the specific details in need of clarification. A number of grammatical issues exist (often across longer sentences; some issues on lines 8, 209, 226-229, 232, 256) Attempting to reformulate some technical ideas and write with simple, shorter sentences might ease reader comprehension.

Your motivation is clearly described but I am not wholly convinced that smaller, more focused venues are generally worse. While they may be less naturally suited for interdisciplinary work they also (intuitively, to me) increase the chance that reviewers and researchers come across work more connected to the areas they are working in.

While you claim there are only mild assumptions of order separability and consistency there appear to be a number of modeling assumptions that may not be realistic. You say that each author must serve as a reviewer, which (as you do mention) is not possible to guarantee. You also say that CoBRA "takes as input only the preference ranking of each author over individual potential reviewers" (line 58/59). Perhaps this is common phrasing in for reviewer assignment work but *as phrased* this seems like an extremely strong requirement. Being more clear that you probably mean something like a similarity score would be an improvement.

**Questions:**

Can you clarify what the X/Y matching is that you are referring to in the Datasets section? Does this indicate that you are giving as input to the algorithm only X out of Y papers from a conference?


How large do the deviating groups for TPMS and PR4A tend to be? If they are extremely tiny or nearly the entire group I would expect that to be a less meaningful deviation than if they are some distinct subfield of research within the conference.

**Limitations:**

The authors have done a very good job of discussing their limitations.

---

> ### Author Rebuttal · Authors · 2023-08-09
>
>
> We thank you for your useful feedback. We will incorporate all your suggestions in our revision. Please see our common answer to all reviewers with regards to your comment about the benefits of focused venues (which we fully agree with).
>
> As for the inputs, we meant to say that we do not *need* numerical similarity scores, only ordinal comparisons (which can always be induced if you have access to similarity scores). But we’ll rephrase this for clarity.
>
> ### **Q1: X/Y matching**
>
> Yes, this is what we mean. We (like Xu et al. [25] and Dhull et al. [26]) are forced to do this because the datasets are anonymized whereas we need to connect the author set to the reviewer set. Thus, we use the conflict information to deduce authorship as best as we can, which is not a perfect process.
>
> ### **Q2: Size of deviating groups**
>
> We agree that the size of the deviating groups is an interesting measure, which we did not consider. In the table below, you can see the *maximum size* of a successfully deviating coalition, averaged across 100 runs, together with the standard error. Recall that each run is a subsampled dataset of size 100, so these can be interpreted as percentages.
>
> It seems that under both TPMS and PR4A across all three datasets, the largest deviating communities are 6-15% of the conference size, which we believe can indeed reflect the sizes of some of the largest subcommunities at CVPR and ICLR. (Note that these are the largest deviations and there are smaller deviations too.)
>
> We will be happy to include this in our revision.
>
>
> | Dataset  | TPMS      | PR4A      |
> |----------|-----------|-----------|
> | CVPR '17 | 6.64±0.77 | 7.5±0.77  |
> | CVPR '18 | 10.54±1.29| 11.49±1.49|
> | ICLR '18 | 11.25±1.76| 15.01±1.76|

---

> > ### Comment · Reviewer_wnxf · 2023-08-17
> >
> > Thank you for the response to my questions. I do agree that groups of 6-15% of the conference size (or even slightly smaller) can certainly represent coherent subcommunities. I believe some mention of this would be a useful addition to the paper. While the paper is compelling I will leave my review as is based on the limitations that CoBRA must overcome before seeing practical usage.

---

### Official Review · Reviewer_HKZ7 · 2023-07-06

**Soundness:** 3 good
**Presentation:** 4 excellent
**Contribution:** 3 good
**Rating:** 7
**Confidence:** 3

**Summary:**

The authors consider a problem of finding reviewer assignments for conference peer review. Specifically, they aim to find a valid reviewer assignment subject to the constraint that no group of authors can achieve a (strictly) preferred reviewer assignment among themselves and a subset of their authored papers. This can be seen as both a fairness constraint and an incentive-compatibility constraint. The authors provide an algorithm to find such a reviewer assignment and theoretically prove that the returned assignment satisfies the desired properties. They then empirically compare the proposed algorithm to several baselines on subsampled real conference data, and show that it achieves reasonable total welfare while other baselines often violate the core constraint.

**Strengths:**

- The problem of finding a paper assignment in the core within the peer review context is interesting and new (to my knowledge). I think the core constraint is well-motivated as a group fairness constraint and is substantively different from the other notions of fairness considered in prior work.
- The writing is generally very clear and the paper is easy to read.
- The author preference model used by the paper is significantly more general than the standard additive utility assumed in other works.
- The empirical results that standard TPMS assignments have significant numbers of core violations on real conference datasets are of practical interest.


**Weaknesses:**

- The setting assumed by the paper is highly simplified compared to the standard peer review setting: all papers are authored by a single author, and no conflicts-of-interest other than authorship are considered. This hinders the practical relevance of the algorithm.
- While the proposed algorithm is claimed to be polynomial-time (Theorem 1), this claim does not seem to be proved and no analysis of the time complexity is provided.
- The empirical results are constructed by subsampling small sets of reviewers and papers. While this is done for computational reasons when evaluating core violations, I’m not sure why the social welfare results were also approximated.


**Questions:**

- Could the authors clarify the time complexity of the algorithm?
- I assume that the USW results in Table 1 are average assigned similarity, not total similarity as defined.

**Limitations:**

The authors clearly stated the limitations of their algorithm, noting that their contribution was primarily conceptual and not practical.

---

> ### Author Rebuttal · Authors · 2023-08-09
>
> We thank you for your useful feedback.
>
> ### **Regarding subsampling**
>
> Indeed, we subsampled sets of reviewers and papers throughout our experiments for consistency (as you noted, this was required for evaluating core violations). If you are interested, the results for USW and ESW without any subsampling are in the table below. It is comforting to note that the qualitative relationships between the different algorithms according to each metric remains the same. It is worth noting that USW goes up across the board while ESW goes down for CoBRA and TPMS but up for PR4A in this case as opposed to the subsampling case from the submission. Note that theoretically, each of USW and ESW can go either up or down when we consider a superset of the data.
>
> We would be happy to add this table to the appendix in our revision, if you wish us to.
>
>
> | Dataset | Algo    | USW   | ESW   |
> |---------|---------|-------|-------|
> |         | CoBRA   | 1.644 | 0.000 |
> | CVPR'17 | TPMS    | 1.970 | 0.000 |
> |         | PR4A    | 1.919 | 0.384 |
> |---------|---------|-------|-------|
> |         | CoBRA   | 1.208 | 0.000 |
> | CVPR'18 | TPMS    | 1.586 | 0.004 |
> |         | PR4A    | 1.560 | 0.731 |
> |---------|---------|-------|-------|
> |         | CoBRA   | 0.251 | 0.015 |
> | ICLR'18 | TPMS    | 0.284 | 0.038 |
> |         | PR4A    | 0.278 | 0.086 |
>
> ### **Q1: Time complexity**
>
> Please see our common response to all the reviewers for the time complexity.
>
> ### **Q2: USW**
>
> Yes, we apologize; these are averaged assigned similarities. We used average rather than sum to report USW and ESW on the same scale for better comparison. We will revise the text accordingly.

---

> > ### Comment · Reviewer_HKZ7 · 2023-08-13
> > **Response to Authors**
> >
> > Thank you for your thorough response and clarifications. I do think it's useful to include the welfare results on the full dataset, or at least to acknowledge that the ranking of the algorithms is not affected by the subsampling.
> >
> > After reading the other reviews and the author response, I will revise my score from a 6 to a 7: despite the practical limitations of the work (most importantly the single-author assumption), I see the conceptual contribution as interesting and valuable.

---

> > > ### Author Response · Authors · 2023-08-14
> > >
> > > Thank you for taking the time to read through our response and revising your score. We will do both: acknowledge the fact that the ranking of the algorithms is not affected by subsampling in the main text and include the welfare results for the full dataset in the appendix.

---

### Official Review · Reviewer_E8Ux · 2023-07-07

**Soundness:** 4 excellent
**Presentation:** 4 excellent
**Contribution:** 3 good
**Rating:** 7
**Confidence:** 3

**Summary:**

The authors propose to use core concept in the context of peer review system. Potentially decreasing an overall welfare, the new paradigm offers a fairer treatment of small sub-communities, erasing an incentive to create an independent venue. The paper provides an algorithm for assigning the reviewers on a restrictive case of single author submission. Experimental support their claims.

**Strengths:**

The authors are trying to address a crucial problem in the modern era ML community---how to assign papers fairly and keep more communities happy. Overall I enjoyed reading the paper, found the notion of core interesting and the proposed methodology sound.

**Weaknesses:**

The authors are pretty up-front with listing the weaknesses of the work. The major one, of course, being the single-author case.

**Questions:**

It would be nice if the authors could write a more general model and definition (not only single author) for future references, while then saying that for now they only treat a particular case as a proof of concept.

**Limitations:**

Section 5 on limitations is pretty clearly written and the list seems exhaustive.

Having purely technical background, I do not feel qualified to comment on the main premise of the work: make smaller communities stay within gigantic conferences. At least in TCS and theoretical ML there are some great smaller conferences (SoCG, SOSA, COLT, ALT, FaccT to name a few). I am not sure that these communities would benefit from being incentivised to stay within say NeurIPS. In particular, it seems to me that fairness and accountability community has greatly benefited from the creation of FaccT (the most recent example I know). So, it would be nice to hear an opinion from people that submit regularly to smaller conferences.

That being said, I am positive about this submission as it tries to tackle a rather complicated, ill posed and timely problem.

---

> ### Author Rebuttal · Authors · 2023-08-09
>
> We thank you for your useful feedback. As per your suggestion, we will define the core more generally in our revision for the benefit of future work, especially given that it is indeed a concept that applies quite broadly.
>
> Please also see our common response to all the reviewers regarding your comment on the advantages of having small conferences.

---

### Official Review · Reviewer_YGAX · 2023-07-25

**Soundness:** 2 fair
**Presentation:** 3 good
**Contribution:** 2 fair
**Rating:** 5
**Confidence:** 4

**Summary:**

This paper proposed an approach that lets the assignment of the peer review model satisfy the core, a fairness requirement over groups of authors, so as to prevent small research communities from having the incentive to deviate and set up their own separate conferences. Through theoretical analysis, the authors found that the proposed method, CoBRA, would return a valid assignment in the core if agent preferences are order separable and consistent. Experimental results show that the proposed CoBRA can produce much fair assignments compared to the baseline approaches.

**Strengths:**

1. This paper addressed an important problem, ensuring group fairness in the peer review process.
2. The proposed CoBRA is theoretically and technically sound.
3. As a theoretical paper, it is not difficult to follow. Several examples and cases are given for the audience to better understand the theoretical analysis as well as the proposed models.

**Weaknesses:**

1. The necessity of achieving fairness in peer review has not yet been well motivated. It is unclear why achieving group fairness will result in the improvement of satisfaction for various communities.
2. The proof of Theorem 1 is not complete. Lemma 1 only validates CoBRA can return the valid assignment in the core but the theoretical analysis on the time complexity of CoBRA to generate the assignment is not complete.
3. The optimality of the valid assignment is not proven.
4. The flexibility of putting the CoBRA in practical use has not been fully discussed. I suggest the authors discuss how CoBRA could deal with the imbalance submission when putting in practical use.
5. It is not easy for non-expert audience to understand this paper, especially why satisfying core is good enough for the review assignments.

Some minor suggestions:
1. The proposed methods can be extended to scheduling tasks in many other application scenarios such as task scheduling in collaborative edge computing, federated learning, etc. Merely producing a fair per-review assignment may limit the contributions of the CoBRA.
2. The technical challenges can move to the introduction sections.

**Questions:**

1. How optimal is CoBRA's assignment?
2. How does each author's paper submission amounts affect the fair peer review assignment made by CoBRA? Some areas will receive a huge number of submissions but the other areas may not. The number of submissions by different authors also varied a lot. How does CoBRA address such data imbalance issues and guarantee group fairness?
3. How to determine groups in the experiment?

**Limitations:**

The limitations have been adequately discussed.

---

> ### Author Rebuttal · Authors · 2023-08-09
>
> We thank you for your review. Please see our common answer to all reviewers for motivations behind the core as well as the exact running time of CoBRA.
>
> ### **Q1 & Q3 (and related comments)**
>
> > How optimal is CoBRA's assignment?
>
> > The optimality of the valid assignment is not proven.
>
> > How to determine groups in the experiment?
>
> We believe that these comments may have stemmed from a fundamental misunderstanding regarding our objective, the core.
>
> At its heart, the core is a *qualitative* notion of fairness; a reviewing assignment is either *in the core* or *not in the core*. There is no notion of “optimality”. One can say that approximation of the core turns this into a quantitative objective, but we prove that our algorithm finds an assignment *in the core* (i.e., achieves the best possible 1-approximation of the core). Thus, there is no optimality to be proven.
>
> Also, the main benefit of the core is that it simultaneously provides a fairness guarantee for *every possible group* (that is, all $2^n-1$ non-empty subset of agents). As such, unlike most definitions of fairness considered in the machine learning literature, where one must specify groups in advance and fairness can only be achieved with respect to those groups, finding a reviewing assignment in the core achieves fairness with respect to all possible groups simultaneously. This also eliminates the need for determining groups in the experiments.
>
> ### **Q2**
>
> > Some areas will receive a huge number of submissions but the other areas may not. The number of submissions by different authors also varied a lot. How does CoBRA address such data imbalance issues and guarantee group fairness?
>
> This is again addressed by the definition of the core, which the assignment returned by CoBRA satisfies. The fairness guarantee offered by the core to every group depends on what the group can achieve on its own. As such, a group that consists of very few people but generates a large number of submissions may not be offered a strong guarantee because such a group, on its own, would not be able to generate high-quality reviews for so many submissions of its own. On the other hand, a group which produces only as many submissions as it can review well on its own and has the internal expertise to produce high-quality reviews for these submissions will find that CoBRA’s assignment treats it quite well, no worse than the satisfactory reviewing outcome it can produce on its own.
>
> We find that this is a natural way in which the core handles imbalances between individuals or research areas without relying on any free parameters (deciding which can become quite controversial in practice).

---

> > ### Comment · Reviewer_YGAX · 2023-08-12
> >
> > 1. CoBRA can only produce assignments in the core. Are those assignments good enough for practical use? I don't think the authors discuss this point. It is not easy for the nonexpert audience to understand why satisfying the core is enough for making the assignment.
> > 2. The authors claim that "a group that consists of very few people but generates a large number of submissions may not be offered a strong guarantee because such a group, on its own, would not be able to generate high-quality reviews for so many submissions of its own". However, this is the situation that a conference may face and cannot be handled by CoBRA because "the fairness guarantee offered by the core to every group depends on what the group can achieve on its own".

---

> > > ### Author Response · Authors · 2023-08-14
> > >
> > > Thank you for putting in further thought and effort in our paper. We appreciate your comments. Our view on these issues is as follows.
> > >
> > > **Regarding comment 1:** Being "good for practical use" has two parts. Is it necessary in practice? Is it sufficient in practice? For the first part (necessity), we provide motivations for needing the core in practice in the submission, and elaborate on it further in our common response. For the second part (sufficiency), we believe that one should look towards the algorithm, not the core, which is only a minimum requirement. An algorithm can have aspects other than satisfying the core which can lead to reasonable assignments in practice.
> > >
> > > This is why we conducted experiments with real data to test CoBRA against state-of-the-art algorithms (TPMS and PR4A) on core and social welfare metrics. We find that CoBRA provides an interesting tradeoff: while CoBRA suffers from a small but not insignificant welfare loss, TPMS and PR4A incentivize realistic deviations (see also our response to Reviewer wnxf regarding the sizes of such coalitions), which CoBRA prevents. In that sense, if TPMS/PR4A are on one end of a spectrum which optimizes welfare, CoBRA is on the other end which focuses on fairness.
> > >
> > > As the very first paper on the subject, we certainly do not claim that CoBRA is ready for practical deployment. We hope that future work can build on ours to design better algorithms that strike a balance between the two extremes to find assignments with better welfare while preventing most, if not all, realistic deviations.
> > >
> > > We hope that answers your question.
> > >
> > > **Regarding comment 2:** We fully agree that a conference may include a community that is small but generates a disproportionately large number of submissions. To be clear, we are not saying that CoBRA will necessarily treat such a community poorly, only that it cannot *guarantee* good treatment to such a community even in the worst case regardless of the problem instance. But no algorithm can do so because, for example, the problem instance may consist solely of such communities and no assignment treating them all well may be feasible (due to the shortage of reviewers). Once again, we hope that future work can build on ours to find instance-dependent bounds, where better guarantees for such communities with reviewing deficits can be provided in cases where other communities exist which contribute reviewing surpluses.

---

> > > > ### Comment · Reviewer_YGAX · 2023-08-21
> > > >
> > > > Thank you very much for the rebuttal. I appreciate that the CoBRA is theoretically and technically sound. When the assignments satisfy the core, group-level fairness will be reached for the possible group identified by CoBRA. The problem is that CoBRA works under the assumption that the identified group should have the capacity to handle the number of submissions on their own, which is too ideal. The reality is the number of submissions is far more than the number of qualified reviewers who can deliver high-quality review comments. Many authors will submit more than one paper, and some may even submit ten papers to the top conference like NeurIPS, CVPR, AAAI, etc. Some hot topic areas will receive a huge number of submissions but some may just have a few. Since the research areas are different, assigning the researchers from the areas will a few submissions to review the papers in the hot topic areas hardly produce high-quality reviews. Such imbalance issues cause the fairness problem of the assignment and CoBRA may fail to handle it. That's why I am worried about CoBRA's flexibility in practical use. I suggest the authors include a detailed discussion in the revised paper. Overall, I believe it is a good paper and I am willing to raise my score to 5. Thank you very much.

---

### Official Review · Reviewer_VPuM · 2023-07-31

**Soundness:** 3 good
**Presentation:** 2 fair
**Contribution:** 3 good
**Rating:** 6
**Confidence:** 4

**Summary:**

The authors frame their investigation into group fairness in the peer review setting in the context of large conferences (i.e. NeurIPS, AAAI) by considering a simplified peer review model that enforces the existence of a valid reviewer assignment.

Within this framework, the authors apply the fairness notion of "the core" to this setting and present an efficient graph-based algorithm that always returns a valid assignment in the core under minor conditions based on author preferences.

The authors empirically validate their method using real data from CVPR and ICLR, and evaluate the cost of utilizing this algorithm (in terms of lost utilitarian and egalitarian welfare) in order to satisfy the fairness notion of "the core" and prevent the incentive for any community to establish their own separate smaller conference (which can lead to research topic insularity and harm interdisciplinary research areas).

**Strengths:**

- While there is a good bit of notation in the model presented, it is kept fairly simple and is clearly described/understandable (to me). This is significant because the paper's primary contributions are methodological and theoretical.

- The authors do well in presenting mathematical definitions will more intuitive descriptions, lowering the cognitive barrier on the reader.

- The proof provided for main theoretical finding (Theorem 1, along with subsequent lemmas and prop) is very rigorous, and thorough.

- Toy example illustrating the execution of the algorithm introduced by the authors (CoBRA) is helpful for building intuition of the method (adding a graph-based visualization of the two companion graphs would be a bonus).

- The empirical results and interpretation are well-described in words and may benefit from modified format to enabled more direct comparison between methods and highlight potential performance trade-offs (please refer to suggestion in "Weaknesses" section on elaboration).

- The authors humbly acknowledge the limitations of their work before it is ready to be used in practice and highlight some potential follow-up directions for investigation.

- Overall comment: much of the content is clearly described, but the presentation structure adds cognitive overhead on the reader and makes it more challenging to understand and utilize the provided information. There are clear strengths in this category, but also real room for improvement. [I struggled to determine the rating in this category, perceived to be between a 2-3 overall. Please refer to "Weaknesses" and "Questions" for more details.]

**Weaknesses:**

Primarily, feedback is around presentation and experimental results/details. Some of these are more significant than others. Overall, the paper is fairly well-written but appears very cramped; the structure and presentation could be improved to support understandability and readability.

- Additional details on empirical setup and experiments is necessary to support reproducibility. (Note that there is a line in the Appendix outlining the computation resources and some core code files are included in the supplementary.)

- Aligned with the comment under "Strengths", there would be benefit in modifying/better aligning the quanititative results with the discussion/explanation (in Section 4 -> Results). The content is good, but presentation is a little hard for me to follow and has room for improvement (perhaps by highlighting best performance for each measure/column in Table 1?). When page limit is not as tight, it'd be helpful to reorganize Section 4 into subsections (and similarly in Section 4).

- It's worth noting that _what_ the authors present is fairly clear, but a clearer motivation of the open problem they are working to address is lacking. (Follow-up question in "Questions" section.)

**Questions:**

1. You mention that theoretically, the CoBRA algorithm is efficient (polynomial time). In terms of actual compute time, how does this scale as a function of the dataset size? Can you provide estimates for the wallclock compute time for the empirical analysis?

2. Why is the introduction of the notion of the core meaningful for the peer reviewer assignment problem space? The Related Work included highlights some relevant work in this area, but does not (from my perspective) very clearly or succinctly motivate the open problem you're addressing in a compelling way. This makes it challenging to determine the potential impact of this work.

3. [A "nice to have" suggestion around presentation] While _technically_ the mathematical definitions of measures computed in empirical evaluation are included in Section 4, they are buried within the main text body; these may be easily missed by the reader or make it harder to understand the paper (it appears that main paper real estate was scarce). I suggest that the authors make room in the main paper to describe the measures used mathematically and verbally (at least prior to publishing) and/or expand upon these along with additional rationale/significance in the Appendix.

**Limitations:**

Practical limitations are well outlined in the final section of the paper, along with directions for follow-up work to build upon this theory-forward paper.

---

> ### Author Rebuttal · Authors · 2023-08-09
>
> We would like to thank you for providing useful recommendations for improving the readability of our paper. We will incorporate them in our revision.
>
> ### **Q1: Time complexity**
>
> The worst-case time complexity of CoBRA is $O(n^3)$ and we have provided the average-case runtime in the PDF attached to the common response. Please see our common answer to all the reviewers for more details.
>
> ### **Q2: Motivation behind the core in peer review**
>
> Please see our common answer to all reviewers.

---

### Author Rebuttal · Authors · 2023-08-09


We thank all the reviewers for their effort and for providing helpful reviews. We will be happy to incorporate all the suggestions of the reviewers as explained in more detail in the individual responses. Let us address two comments raised by multiple reviewers in this common response.

### **Motivation behind the core in peer review**

There are three key motivations for studying the core in the peer review setting.

1) **Fairness:** The core acts as a notion of group fairness, which provides a guarantee that every possible group (even a community that doesn’t yet have a well-established identity) will be treated well relative to what the group can achieve on its own, which is a function of the reviewing burden imposed by the group versus the reviewing capacity contributed by the group.

2) **Stability:** The core also acts as a notion of stability because it ensures that no group would have an incentive to break off due to a feeling of being mistreated by the large conference (such as NeurIPS). Note that communities may still prefer to set up their specialized conferences for very good reasons including those outlined by the reviewers, but we believe that receiving low-quality reviews should not be one of them. This is precisely what the core aims to ensure.

   We also remark that while specialized conferences have their advantages, so do large conferences. For example, they can find the diverse reviewing expertise needed for emerging multidisciplinary areas and provide a venue for interdisciplinary dialogues to take place. Thus, it is useful to retain at least some large conferences alongside specialized conferences, but this would be difficult if various communities keep breaking off due to not receiving high-quality reviews.

   In this sense, we don’t see the core as creating roadblocks to the creation of specialized conferences, but rather as a way of mitigating harm imposed on communities in large conferences, thereby maintaining their existence alongside more specialized venues.

3) **Robustness:** A key benefit of the core is that it is a robust definition of fairness that does not require specifying groups in advance, unlike most definitions studied in the machine learning literature. Instead, it simultaneously provides a fairness guarantee for *every possible group* (that is, all $2^n-1$ non-empty subset of agents); this includes groups defined based on sensitive attributes, intersectional groups, or even groups that do not yet have a well-established identity. Further, the guarantee scales in a principled manner, without having to set any controversial parameter values, as a function of what each group can achieve on its own, as mentioned above.

We will clarify these more explicitly in our revision and revise language in our paper which, unfortunately, seems to incorrectly suggest that “preventing” small communities from deviating is our intention (it is not!).

### **Running time of CoBRA**

We apologize for not including a running time analysis of CoBRA. Since we show that the number of assigned reviews grows monotonically (with each increase evidently requiring polynomially many steps), we believed it was immediate that the algorithm runs in polynomial time, which is all that Theorem 1 claims. Like many theoretical works, we were not overly concerned with the exact time complexity. However, we see that it is useful to not only lay out the polynomial time argument, but to also identify the exact complexity so future works can perhaps improve on it. We do so here and will include this in our revision.

First, let us consider the time complexity of PRA-TTC. In each iteration, the algorithm assigns at least one extra reviewer to at least one incompletely-assigned submission. This can continue for at most $m \cdot k_p \leq n*k_a$ iterations, where the inequality follows from our condition for ensuring the existence of a feasible reviewing assignment. In each iteration, it takes $O(n)$ time to find and eliminate a cycle in the preference graph and $O(n^2)$ time to update the preference graph (for an arbitrarily-picked incompletely-assigned submission of each agent, we need to find the most qualified reviewer who can be additionally assigned to it). Thus, in total, the runtime of PRA-TTC is $O(n^3)$ because $k_a$ is a small constant in practice.

After PRA-TTC terminates, CoBRA calls the Filling-Gaps algorithm. However, Lemma 3 ensures that at the end of PRA-TTC, $|L \cup U| \le k_p+1$, which is a small constant. And Filling-Gaps only makes local changes that affect these constantly many agents. As such, the running time of Filling-Gaps is constant as well.

Overall, the time complexity of CoBRA is $O(n^3)$.

For the average-case runtime, we have attached a PDF to the common response, which shows the running time of CoBRA as a function of the number of submissions in the conference. Results are obtained by subsampling submissions from the three datasets using the same process as in our experiments section, and average runtime over 25 runs is shown together with the standard error. Across all datasets, CoBRA runs in less than half a minute even with 800 submissions.

We will be happy to include this figure in the appendix in our revision.

---

### Decision · Program_Chairs · 2023-09-21

**Decision:**

Accept (spotlight)

**Comment:**

The paper defines a group fairness notion for the peer review assignment problem using a concept of core from economics where the goal is to find an assignment that treats every possible group as well as they could be on their own. The paper presents an efficient (polynomial time) graph-based algorithm that always returns a valid assignment under certain conditions on author preferences. The reviewers appreciate the timeliness and significance of the problem, as well as the soundness of the algorithm and the rigor with which the theoretical and empirical results are presented. The problem statement, the novel notion of fairness and the presented solution makes this a strong paper for NeurIPS. However, the reviewers noted a few things that I would encourage the authors to add to the final version. First, the submission should include a time complexity analysis of the solution (even if it can be easily inferred). Second, clarify the misunderstanding that the goal of the fairness notion is not for the paper authors to not breakaway to form a special conference, but for them to get good reviews from relevant reviewers. Finally, the authors should clearly state other practical limitations of the approach that arise due to the assumptions. This will ensure that future work on open questions and allow for adoption of better peer-reviewing assignment algorithms or practices.